# Continuous Multi-step Predictions of Highly Imbalanced Multivariate Time Series via Deep Learning Network

## Abstract

Multi-step prediction of multivariate time series has always been a very popular research topic across industries. We focus on the scenario in which the data with severe imbalance problem caused by the 0 expansion in regression analysis, and meanwhile the data contains complex textual information. Such data is very common in customer's life time value evaluation tasks in businesses. The commonly used two-stage modeling scheme effectively predicts whether or not a customer will pay for a product or service at the next moment. However, it is incapable of continuously forecasting potential payment values due to the strong imbalanced and randomness distribution of the data. In this paper, we propose a feature learning based deep learning method for imbalanced multivariate time series (FLIMTS). The innovative use of a weighted quantile loss in our proposed method handles the highly imbalance problem in regression. Furthermore, FLIMTS incorporates both the customer's payment sequence and the behavioral characteristics of their interests which allows for more accurate predictions. Empirical analysis shows that FLIMTS has significant advantages and performs better than the existing two-stage approaches on common model evaluation criteria.

**Keywords**: highly imbalanced data, multivariate time series, LTV study, feature learning

## 1 Introduction

The Multi-step forecasting for multivariate time series has been widely used in daily life, such as finance, medicine, meteorology and, and other fields with very great commercial values. With the rapid development of computer science, data structure changes significantly. Multivariate time series contain complex textual information gradually becomes more common. Such complex multivariate time series data is usually accompanied with severe data imbalance problem of 0 expansions, which makes the multi-step predictions extremely difficult. We are interested in solving such multi-step ahead prediction problem of the informative multivariate time series data with severe imbalance problems in the data structure.

The motivation of our research is from the statistical modeling and multi-step prediction of the customer's life time value (LTV) sequence in commercial activities. We aim to study the impact of the user's payment habits on the payment amount. The customer's LTV refers to the total profit made from users during the time from product query to terminating the online services. As an important business indicator, the predicting the user's payment values is a critical business requirement. Predicting the user's payment directly determines the service providers' revenue capacity and their online service quality. The user's payment value data is a typical extremely imbalanced multivariate time series. The highly imbalanced distribution of the response variable $y_t$, which refers to the user's payment label value at the time $t$, brings extreme difficulties to provide continuously multi-step ahead predictions of the multivariate time series data. This problem has been an unsolved challenge in industry for a long time.

In our scenario, the LTV prediction is a typical imbalanced regression analysis problem since the user's payment value follows a highly imbalanced distribution with zero inflated. The proposed method should be conducted with the imbalanced learning technique. However, the available solu-

tions for the imbalanced problem are mostly designed for classification purposes. Even though Yang et al. (2021) has made some breakthroughs, the imbalance problem in regression is still a challenge question, and only a few achievements have been made in the related fields so far. Basically, there are two approaches to alleviate the imbalanced problem for classification, the data-based method and the model-based method. For the data-based method, undersampling in the majority class (Chawla et al., 2002; Han et al., 2005; He et al., 2008; Douzas and Bacao, 2019), and oversampling in the minority class (Chawla et al., 2002; Han et al., 2005; He et al., 2008; Douzas and Bacao, 2019) were used. However, these methods are not applicable to the regression problem directly, since the resampling method will bring strongly inductive bias to the distribution of the continuous label values. Compared to the data-based methods, the model-based methods might be applicable to the regression problems. For example, Lin et al. (2017); Li et al. (2019; 2020) added weights to samples or adjusted the objective (loss) function of the model. Yin et al. (2019); Huang et al. (2016); Yang and Xu (2020); Shu et al. (2019) used methods such as transfer learning, metric learning, and meta-learning techniques. Currently, the most competitive method for the imbalanced learning is by Kang et al. (2019). They decoupled the imbalanced learning into two stages of normal sampling in the feature learning stage, and balanced sampling in the Label Learning stage. The decoupled strategy achieves optimal modeling results so far.

Since the prediction of user's payment values involves both regression and classification, the most commonly used solution in the industry is a two-stage approach, where the prediction process is disassembled into two sub-tasks: the classification task (stage-I: whether the user pays) and the regression task (stage-II: the payment amount of paying users) (Vanderveld et al., 2016; Chamberlain et al., 2017; Wang et al., 2019). Machine learning algorithms are adopted in these two tasks. Suppose the prediction result of the classification task is $\hat{p}_i$, and the given threshold is $\omega$, and the prediction result of the regression task is $\hat{v}_i$, then the model of the predicted paid value is $\hat{y}_i = I(\hat{p}_i > \omega)\,\hat{v}_i$. In the industry, the LightGBM algorithm (Ke et al., 2017) is commonly adopted for engineering implementation. We name this type of method as 2Stage-LGBM algorithm. The downside of these two-stage algorithms is that the model only can provide one time step ahead prediction due to the imbalance problem of the data. The current two-stage approaches achieve the multi-step ahead predictions in a clumsy way of that a series of independent predictive models for each target moment are established. This independent modeling scheme is not only difficult to conduct, but also a waste of time and computational resources. It increases the model maintenance cost with a unsatisfactory prediction accuracy. Therefore, we need a more sufficient predictive modeling strategy which allows for continuously multi-step ahead predictions in one model. Meanwhile, since we are analyzing the sequential data, Time should be included into the feature structure of the desired model.

In this article, we propose a deep learning algorithm based on the Feature Learning for imbalanced multivariate time series (FLIMTS). FLIMTS includes two parts: Representation Learning and Label Learning. Compared with the commonly used algorithms in industry, FLIMTS has two major advantages. First, by innovatively introducing a weighted quantile loss, it successfully eliminates the impact of the imbalanced data distributions. Second, FLIMTS fully utilizes the user's portrait features, and deeply analyzes the static features and sequence features of the multivariate sequence data via feature learning processes to obtain more comprehensive sequence feature information. In the empirical analysis, we compare FLIMTS with the 2Stage-LGBM approach on a public data set. The results show that the proposed method performs better than the 2Stage-LGBM algorithm on common model evaluation criteria.

The rest of the article is organized as follows. Section 2 introduces the model and the details of the proposed algorithm. Section 3 is the empirical analysis. We compare our proposed algorithm with the 2Stage-LGBM algorithm on two data sets. Discussion in Section 4 concludes the article.

## 2 METHODOLOGY

### 2.1 MODEL AND NOTATION

The proposed method is a multi-step forecasting deep learning algorithm based on the feature learning for multivariate time series with heavy imbalance problem. Suppose that at time $t$, $\mathbf{y}_t^{(q)}$ are independent $q$ step observations generated from the following imbalanced multivariate time series model:

$$\mathbf{y}_t^{(q)} = \mathcal{F}(\mathbf{x}_{t-p}, \ldots, \mathbf{x}_{t-1}, \mathbf{x}_t, \mathbf{y}_t^{(p)} | \Theta) + \boldsymbol{\epsilon}_t^{(q)}, \tag{1}$$

where $\boldsymbol{\epsilon}_t^{(q)}$ is the random error, $\mathcal{F}$ is a unknown nonlinear mapping function, and $\Theta = \{\Theta_{rep}, \Theta_{lab}\}$ is the parameter set of the Representation Learning module and the Label Learning module. $p + 1$ is the size of the window of the previous data series used for later data series prediction, and $q$ is the length of the predicted series in the multi-step prediction. $\mathbf{y}_t^{(p)} = [y_{t-p} \quad y_{t-p+1} \quad \cdots \quad y_t \quad]^T \in \mathbb{R}^{p+1}$ is the user paid value label sequence from time $t - p$ to $t$. $\mathbf{y}_t^{(q)} = [y_{t+1} \quad y_{t+2} \quad \cdots \quad y_{t+q} \quad]^T \in \mathbb{R}^q$ is the user paid value label sequence from time $t + 1$ to $t + q$. $\mathbf{x}_t \in \mathbb{R}^m$ is the $m$-dimensional independent variable related to the feature variables at time $t$. These features can be grouped into two feature vectors, which are the static feature vector $\mathbf{x}_t^s \in \mathbb{R}^{m_1}$ and sequence feature vector $\mathbf{x}_t^h \in \mathbb{R}^{m_2}$, where $m = m_1 + m_2$. The static features do not change over time. For example, some portrait features, such as gender and age in $\mathbf{x}_t$, can be regarded as static features, where

$$\mathbf{x}_t^s = \mathbf{x}_{t'}^s = \mathbf{x}^s = [x_1^s \quad x_2^s \quad \cdots \quad x_{m_1}^s]^T, \qquad \forall\, t \neq t'.$$

Observations of the sequence features change over time. For example, the number of user logins in and the user's historical payment information $\mathbf{y}_t^{(p)}$ are both sequence features, where

$$\mathbf{x}_t^h = [x_{t,1}^h \quad x_{t,2}^h \quad \cdots \quad x_{t,m'}^h \quad y_{t-p} \quad y_{t-p+1} \quad \cdots \quad y_t]^T = [x_{t,1}^h \quad x_{t,2}^h \quad \cdots \quad x_{t,m_2}^h]^T,$$

where $m_2 = m' + p + 1$. Finally $\mathbf{x}_t$ converts to

$$\mathbf{x}_t = \begin{bmatrix} \mathbf{x}^s \\ \mathbf{x}_t^h \end{bmatrix} = [x_1^s \quad x_2^s \quad \cdots \quad x_{m_1}^s \quad x_{t,1}^h \quad x_{t,2}^h \quad \cdots \quad x_{t,m_2}^h]^T.$$

According to equation 1 we will use the sequence state information (including the static feature information, the sequence feature information, and the label information) of the first $p$ moments of the data sequence to predict the label information of the following $q$ moments of the data sequence.

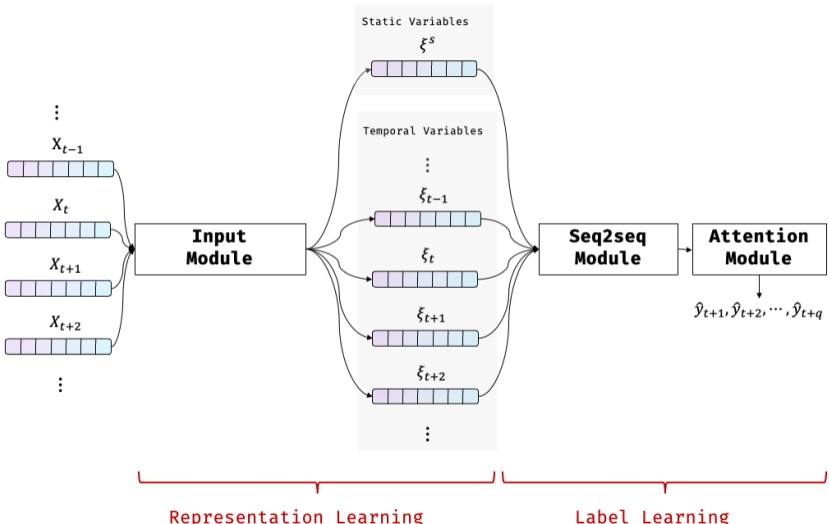

Figure 1: Model Architecture Diagram of FLIMTS

Figure 1 is the computational framework of the FLIMTS algorithm. The key idea of the FLIMTS algorithm is to fit a nonlinear mapping $\mathcal{F}$ using a deep neural network. The algorithm structure includes two parts: the Representation Learning module and the Label Learning Module. In the

Representation Learning part, the deep neural network combines continuous variables and discrete variables to achieve a desired analysis effect. The Representation Learning part is to decompose the independent variable sequence into static features and sequence features, and then convert them into the continuous representation vectors. The Label Learning part includes the Seq2seq module, the Attention module and the the Output module. It maps the continuous representation vectors generated from the Representation Learning to the target value of the response variables. For each sequence data sample, it will be converted into a vector through the Representation Learning module firstly, then will be processed by the Seq2seq module, the Attention module and the output module to obtain the Multi-step prediction values. The computational workflow of the proposed algorithm FLIMTS is summarized in Algorithm 1. The details of our designed deep learning networks are introduced in Appendix A.

---

**Algorithm 1:** FLIMTS

**Input:** $\{\mathbf{x}_\tau\}$, for $\tau \in [t-p, t]$

**Output:** the $q$-step predictions of the sequence lables $\widehat{\mathbf{y}}_t^{(q)} = \left[\hat{y}_{t+1}, \hat{y}_{t+2}, \ldots, \hat{y}_{t+q}\right]^T$

**Step 1 Representation Learning:**

(a) Obtain the embedded static feature vector $\boldsymbol{\xi}^s$ of the input variable through the static feature processing module (Algorithm 2 in Appendix A.1):

$$\boldsymbol{\xi}^s \leftarrow StaInput(\mathbf{x}^s);$$

(b) Obtain the embedded sequence feature vector $\boldsymbol{\xi}_\tau^h$ of the input variable through the sequence feature processing module (Algorithm 3 in Appendix A.1):

$$\boldsymbol{\xi}_\tau^h \leftarrow TempInput(\mathbf{x}_\tau^h), \quad t - p \leq \tau \leq t.$$

**Step 2 Label Learning:**

**Step 2.1** process the feature vector through the Seq2seq module (Algorithm 4 in Appendix A.2.1):

$$\left[\boldsymbol{\hbar}_{t-p} \ldots \boldsymbol{\hbar}_t \ldots \boldsymbol{\hbar}_{t+q}\right]^T \leftarrow Seq2seqModule\left(\left[\boldsymbol{\xi}^s, \xi_{t-p}^h, \ldots, \boldsymbol{\xi}_{t-1}^h, \boldsymbol{\xi}_t^h\right]^T\right);$$

**Step 2.2** predict the $q$-step sequence labels vector $\widehat{\mathbf{y}}_t^{(q)}$ through the Attention module (Algorithm 5 in Appendix A.2.2):

$$\widehat{\mathbf{y}}_t^{(q)} \leftarrow AttentionModule\left(\left[\boldsymbol{\hbar}_{t-p} \ldots \boldsymbol{\hbar}_t, \ldots \boldsymbol{\hbar}_{t+q}\right]^T\right).$$

---

## 2.2 Weighted Quantile Loss and Model Optimization

The proposed deep learning network contains a large number of parameters. We adopt a two-stage parameter optimization strategy for the proposed method, where the parameters of the Representation Learning model and the Label Learning model are optimized separately and sequentially. In the Representation Learning stage, we use the common Quantile Loss function. In the Label Learning stage, to deal with the highly imbalanced problems of the multivariate time series, we innovatively introduce a Weighted Quantile Loss function, which greatly improves the effect of multi-step forecasting. Then we iteratively update all parameters based on the gradient back propagation technique.

The Representation Learning is the first stage of the proposed FLIMTS algorithm, since we do not have to consider the prior information of the label distribution, we only need to learn the pattern of the data distribution. Therefore, we do not have to do subsampling in this modeling stage. Balanced subsampling is enough if necessary. In the learning process, the optimal estimates of all parameters are obtained by minimizing the following Quantile Loss

$$QLoss(\eta, y_i, \hat{y}_i) = max\big\{\eta \cdot (\hat{y}_i - y_i), (1 - \eta) \cdot (y_i - \hat{y}_i)\big\},$$

where $\eta \in (0,1)$ adjusts the prediction tendency of the algorithm. If $\eta$ is large, the algorithm parameters will be trained in the direction of underestimating the label value. If $\eta$ is small, it will be trained in the direction of overestimating the label value. When $\eta$ is set to $0.5$, the effect of the Quantile Loss will be equivalent to the Absolute Value Loss.

The Label Learning phase is the second stage of the proposed FLIMTS algorithm. The purpose of this stage is to fit the mapping relationship between the feature vectors and the label values. Therefore, the prior knowledge of the label distribution will have a significant impact on the modeling process. In the Label Learning phase, parameters are optimized by minimizing the Weighted Quantile Loss. To construct the Weighted Quantile Loss, we need to obtain its asymptotic distribution of the label value. Suppose that the partition of the range of the label value is $x_{(0)} < x_{(2)} < \cdots < x_{(j)} < \cdots < x_{(N)} = \infty$. We estimate the distribution of the label value $F(y)$ by the empirical distribution function $G_N(y) = \frac{1}{N} \sum_{j=1}^{N} I(Y_j \leq y)$. The weight of the $i^{th}$ label value is

$$ w_i = \sum_{j=0}^{N} \frac{I(x_j \leq y_i < x_{j+1})}{\int_{x_j}^{x_{j+1}} f(x)dx}, \ i = 1, \ldots, N. $$

Then the Weighted Quantile Loss is

$$ WQLoss(\eta, Y, \hat{Y}) = \sum_{i=1}^{q} w_i \cdot QLoss(\eta, y_i, \hat{y}_i). $$

The Weighted Quantile Loss function is designed to be sensitive to the skewness of the target label distribution and the sensitivity is reinforced for intervals with few samples, since the total loss coming from these parts are highly likely to be underrated due to its small quantity of samples in the optimization process. In reality, in order to improve programming efficiency and reduce the time expense caused by the memory exchange in the communication between different devices (CPU and GPU), our empirical distribution function $G_N(y)$ is usually obtained by integrating the results of each batch in the parallel computing process. This method may cause small amount of deviation when estimating the distribution of the data with imbalance problem. However, this small deviation will gradually converge as the number of training rounds and the sample size increases. Therefore it will have little impact on the overall fittings of the model.

Let $\Theta = \{\Theta_{rep}, \Theta_{lab}\}$ is the set of parameters that need to be optimized in the proposed model, where $\Theta_{rep}$ is the parameter set of the Representation Learning part, $\Theta_{lab}$ is the parameter set of the Label Learning part. These two sets of the parameters cannot be completely separated in the training process. In the Representation Learning part, $\Theta_{lab}$ is also updated while optimizing $\Theta_{rep}$. Meanwhile, in order to improve the stability of parameter estimation, we need to optimize the collective loss under different values of $\eta$. In our empirical analysis we use $\eta \in \{0.3, 0.5, 0.7\}$. Let $L_{rep}$ be the Quantile Loss of the Representation Learning.

$$ L_{rep} = \sum_{\eta} \sum_{i=1}^{q} QLoss(\eta, \hat{y}_{t+i}.y_{t+i}), $$

$\Theta_{rep}$ is optimized and updated by the minimizer of $L_{rep}$.

$$ \Theta_{rep} = \Theta_{rep}^* - \alpha \frac{\partial L_{rep}}{\partial \Theta_{rep}^*}, \qquad \Theta_{lab(rep)} = \Theta_{lab(rep)}^* - \alpha \frac{\partial L_{rep}}{\partial \Theta_{lab(rep)}^*}, $$

where $\Theta^*$ is the previous state of the parameters, and $\Theta$ is the updated state of the parameters. $\Theta_{lab(rep)}$ is the parameter set of the Label Learning updated in the Representation Learning stage. Let $L_{lab}$ be the Weighted Quantile Loss of the Label Learning.

$$L_{lab} = \sum_{\eta} \sum_{i=1}^{q} WQLoss(\eta, \hat{y}_{t+i}, y_{t+i}).$$

$\Theta_{lab}$ is firstly initialized by $\Theta_{lab(rep)}$, that is $\Theta_{lab}^{(0)} = \Theta_{lab(rep)}$. Then $\Theta_{lab}$ is optimized and updated by

$$\Theta_{lab} = \Theta_{lab}^{*} - \alpha \frac{\partial L_{lab}}{\partial \Theta_{lab}^{*}}.$$

In addition, the generalization performance of the model is improved by controlling the Dropout Rate ($DPR$).

## 3 EMPIRICAL ANALYSIS

We compare the proposed method with the commonly used 2Stage-LGBM algorithm on the public dataset AVSC. The 2Stage-LGBM algorithm is implemented based on the LightGBM algorithm and is one of the best prediction scheme in the industry for multivariate time series with highly imbalanced problems. Notice that, in our experiment we make four step forward predictions from time $t + 1$ to $t + 4$. FLIMTS only needs to build a single model to generate continuous multi-step predictions, while the 2Stage-LGBM has to build four separate prediction models for each target moment.

The AVSC dataset is a desensitized transaction data (Dua and Graff, 2017) from the Acquire Valued Shoppers Challenge. This dataset includes transactions of some brick-and-mortar stores over a period of time with 11 features for each data record, such as User ID, Store ID, Item Category, Item Subcategory, Company ID, Brand ID, Purchase Date, Number of Purchased Item, Measurement Unit of the Purchased Item, and Purchase Amount. The original data structure of the AVSC dataset is not very suitable for the purpose of our analysis since it does not show out the user's consumption behaviors very well. The data must be preprocessed before performing any further analysis. Let's define the user's payment value is the amount of repurchase made by a user who has a purchase record before. We first clean and aggregate the original data based on the customer ID, and obtain the monthly customer's consumption data, which is the overall paid value of each customer in 8 months. The reprocessed data has two groups of features, the Payment features and the Context features. The Payment features are used to describe the customer's payment behavior, including the number of payments, the frequency of payments, and the average amount of each payment, etc. The Context features are the statistical characteristics of existing customers in a store, such as the payment amount per capita and the payment frequency per capita, etc. Since this is a monthly data, the data at each time point is the cumulative purchase value within each month. Not surprisingly, the paid value of is extremely imbalanced, since the ratio of non-paying users to paying users exceeds $3 : 1$. Among these paying users, the number of customers with higher paid value decreases sharply as the paid value increases. To conform the data to the real business scenario, we split the dataset into the training set and the test set based on the dates. We use the data before Dec, $1^{st} 2012$ as the training set, which has 1,355,880 data records. Then the test set size is 356,980. Their ratio is about $8 : 2$.

Since the user payment value is a continuous variable, we choose the mean square error ($MSE$) and the mean absolute error ($MAE$), which are two commonly used model evaluation criteria in regression. In reality, the downstream tasks of paid value prediction are usually related to the ranking of users (such as selecting TOP-N from users for advertising, etc.). Therefore we adopt the $rAUC = P(\hat{y}_1 > \hat{y}_2 \mid y_1 > y_2)$ ($regression - AUC$) criterion, which measures the ranking quality of the regression model. A better model can rank samples with larger true label values ahead of samples with smaller true label values when sorting the samples according to their predicted values from large to small. The larger the $rAUC$ is, the higher the accuracy of the algorithm in sorting users according to the predicted value, and vice versa.

The computational environment of experiments in this article are: $CPU Inter i7 - 10700; GPU Nvidia RTX 3060; RAM 32 Gb; Software Python 3.9 + CUDA 11.1$. We use the

218  open source $SQL$ query engines (Impala and Trino) for data cleaning and feature engineering. The
219  implementation of algorithm engineering relies on the Pytorch (Paszke et al., 2019) environment,
220  and we also uses the Pytorch-lightning module for parameter optimization.

## 3.1  EXPERIMENT RESULTS

222  In the experiment we set the parameters $p = 6, q = 4$ in the model equation 1. The initialization
223  settings of hyperparameters are: the learning rate is 0.0001; the number of LSTM layer is 1; the
224  dimension of the intermediate and hidden variables in the Seq2seq module is $d_k = 128$; the number
225  of heads in the Multi-head Self Attention module is $H = 8$; the dropout rate is $DPR = 0.2$.

226  The results of the model evaluation criteria on the training set are shown in Figure 2. The trend of
227  all criteria drops rapidly in the early stage of training, and tends to be a stable fluctuation later. This
228  indicates that the proposed model converges fast in the training stage.

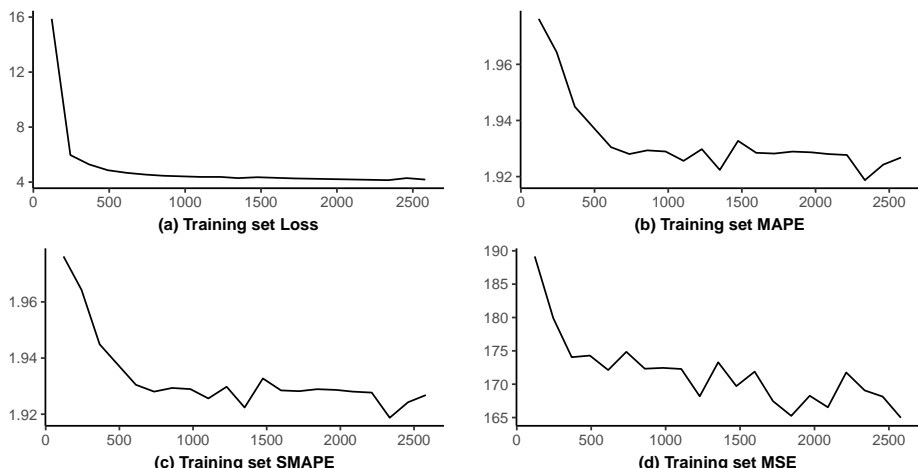

Figure 2: The convergence rate of the proposed model on the training stage.

229  In practice, the predictive quality of those paying users can better reflect the advantages of the
230  models, thus attract more attention in business. We compare the model performance of these two
231  algorithms in predicting the paid value at the $(t + 1)th$ moment on two customer groups, which are
232  the All User group and the Paying User group. The result is summarized in the Table 1. We see that
233  the MSE and MAE of FLIMTS are much smaller than the 2Stage-LGBM approach, which means
234  that the mean and median of the predicted paid value by FLIMTS are closer to the true value than
235  the 2Stage-LGBM algorithm for both All User group and Paying User group. In terms of the rAUC,
236  since both model schemes are based on regression analysis, therefore the rAUC of FLIMTS is very
237  close to the 2Stage-LGBM algorithm, which is a reasonable result.

Table 1: Model Performance Comparison

|            | Algorithm    | MSE    | MAE  | rAUC |
|------------|--------------|--------|------|------|
| All User   | FLIMTS       | **51.54**  | **0.85** | **0.98** |
|            | 2Stage-LGBM  | 151.05 | 0.91 | 0.98 |
| Paying User| FLIMTS       | **208.82** | **1.55** | 0.90 |
|            | 2Stage-LGBM  | 615.32 | 2.18 | **0.90** |

238  Figure 3 shows the MSE, MAE and rAUC of the FLIMTS and the 2Stage-LGBM algorithms in
239  multi-step ahead predictions of the paid values at the next four moments of time $t+1$ to $t+4$. In terms
240  of the rAUC, the proposed algorithm shows a slight advantage over the 2Stage-LGBM algorithm as
241  the size of the prediction time step increases. For the MSE and MAE, the proposed algorithm
242  provides much smaller results than that of the 2Stage-LGBM algorithm, and this advantage grows
243  as the size of the prediction time step increases.

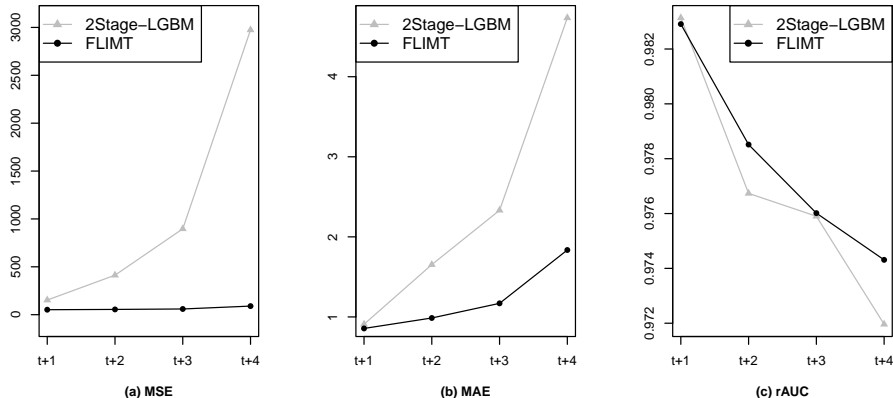

Figure 3: The MSE, MAE and rAUC of different algorithms in multi-step predictions of the paid value at the next four moments of time $t + 1$ to $t + 4$.

Figure 4 shows the multi-step prediction results of the paid value from time $t + 1$ to $t + 4$ based on the FLIMTS algorithm and the 2Stage-LGBM algorithm for four types of paying value users, which are the High paying value users, Mid paying value users, Low paying value users, and Null paying value users. The prediction results at all four moments show that FLIMTS has better predicting performance than the 2Stage-LGBM method for all four types of paying value users with much less cost of the computational resources, since FLIMTS only needs to build a single model to generate multi-step predictions continuously, while the 2Stage-LGBM has to build four separate prediction models for each target moment from $t + 1$ to $t + 4$.

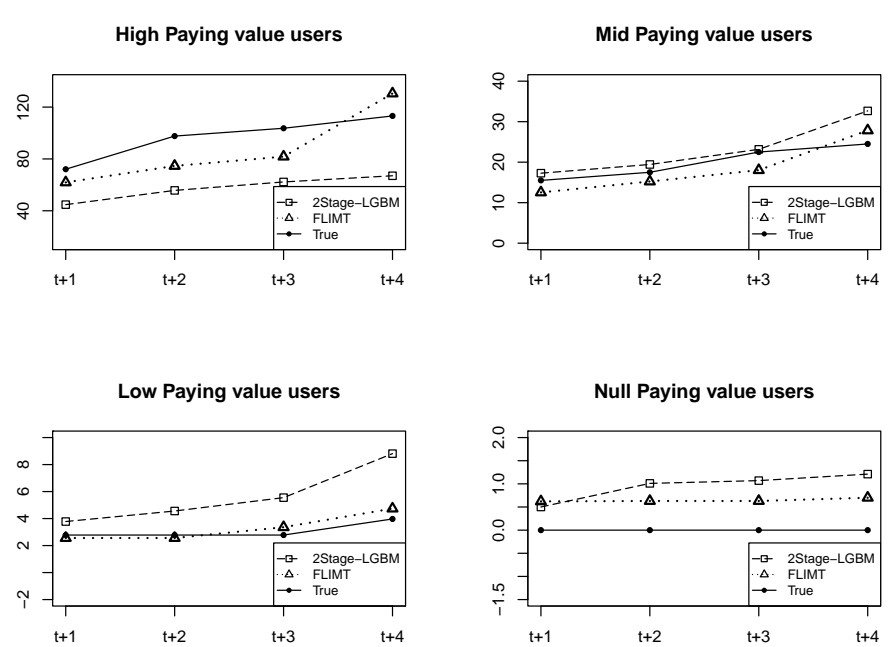

Figure 4: Multi-step predictions of the paid value at the next four moments from $t + 1$ to $t + 4$ for four types of paying value users under two methods.

## 4 CONCLUSION

In this paper we propose a new deep learning algorithm, FLIMTS, for the multi-step forecasting of the multivariate time series with severe data imbalance problem. The great advantage of the FLIMTS algorithm is that by introducing a weighted quantile loss, we greatly reduce the influence of the imbalanced data distribution problem. Therefore, the proposed method only needs to build one predictive model to generate multi-step ahead predictions for a sequence of target moments for imbalanced multivariate time series. In contrast, the traditional 2Stage-LGBM algorithm must build separate predictive models at each target moment to achieve satisfactory multi-step predictions. Additionally, the prediction accuracy is improved by the rigorously designed deep learning networks, which combine the Representation Learning and Label Learning based on the Feature Learning techniques.

AUTHOR CONTRIBUTIONS

ACKNOWLEDGMENTS

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

# A APPENDIX

## A.1 REPRESENTATION LEARNING NETWORK

The Representation Learning is designed to learn the information of all features and convert them into the representation vectors for the further quantitative analysis. At each time $t$, the feature vector of each sample is $\mathbf{x}_t = (x_1^s, x_2^s, \ldots, x_{m_1}^s, x_{t,1}^h, x_{t,2}^h, \ldots, x_{t,m_2}^h)$, where each $x$ represents the observation of the corresponded feature at time $t$. The algorithm takes the sequence features as the input to the Encoder. In the modeling process, since these features are not necessarily numerical continuous, and discrete features cannot be directly used in the deep neural network model, it is necessary to use the embedding method to convert these features into vectors with specified dimension. And this is also the main purpose of using the Representation Learning. In the following content, we use $\boldsymbol{\xi}^s$ and $\boldsymbol{\xi}_t^h$ to represent the static feature vectors and the sequence feature vectors that are learned from the Representation Learning respectively.

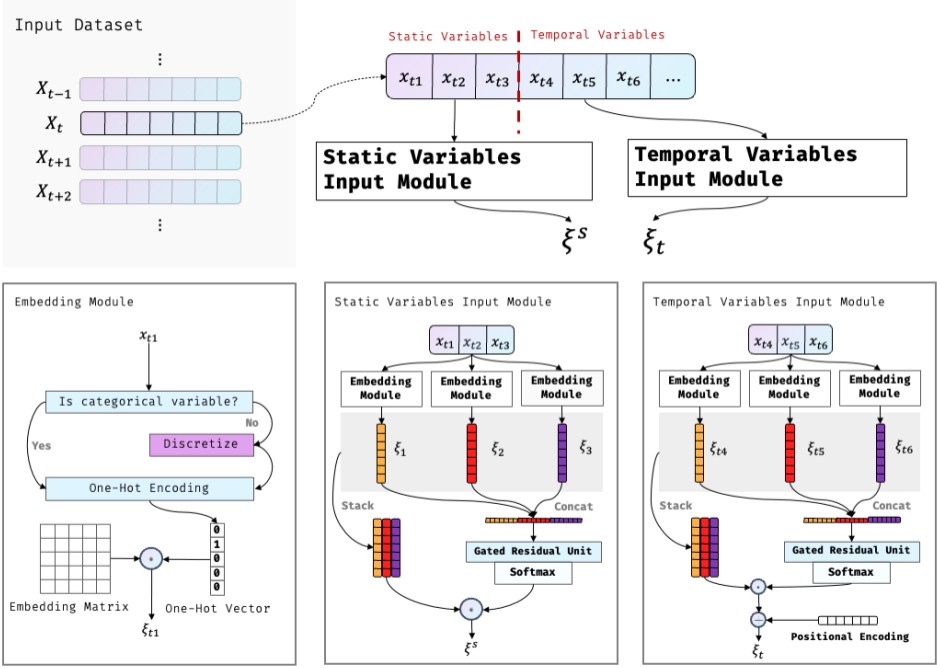

Figure 5: The operational flow chart of the Representation Learning

Suppose we set the embedding vector $\boldsymbol{\xi}_i$ at time $t$ to be a $d_k$ dimensional vector. For the $i^{th}$ discrete feature $X_{t,i}$, we encode the feature firstly, and then find the corresponded $d_k$-dimensional embedding vector $\boldsymbol{\xi}_i$ from the Embedding Matrix of $x_{t,i}$, where $d_k$ is a hyperparameter that has to be decided in advance. For example, in the following empirical analysis section, we set $d_k = 128$. For the $j^{th}$ continuous feature $X_{t,j}$, according to the specific situation, we could transform it into a discrete variable using the binning technique or convert it into a $d_k$ dimensional vector $\boldsymbol{\xi}_j$ through a $d_k$

dimensional fully connected network layer. At this stage, both the parameters of the Embedding Matrix and the fully connected network can be optimized in the training process.

Figure 5 is the operational flow chart of the Representation Learning. At time $t$, the Representation Learning divides the input vector $\mathbf{x}_t$ into the static features and sequence features, and processes them separately. For the static features, we designed a static feature processing algorithm, StaInput, and its computational workflow is described in the Algorithm 2. For the dynamic sequence features, we designed a sequence feature processing algorithm, TempInput, and its computational workflow is described in the Algorithm 3. Both algorithms map each feature into a dense embedding vector via the embedding process,

$$\boldsymbol{\xi}_i^s \leftarrow Emb_i(x_i^s),$$
$$\boldsymbol{\xi}_{t,j}^h \leftarrow Emb_i(x_{t,j}^h),$$

where $i \in [1, m_1], \boldsymbol{\xi}_i^s \in \mathbb{R}^{d_k \times 1}$, and $j \in [1, m_2], \boldsymbol{\xi}_{t,j}^h \in \mathbb{R}^{d_k \times 1}$. Then obtain the representation vectors by the weighted average of these embedding vectors. The difference is the representation learning for the sequence features have an additional temporal information encoding step.

To more accurately extract the representation information of the sequence, the Representation Learning estimates the weight of the embedding vector, and uses the weighted average method to calculate the representation vectors $\boldsymbol{\xi}^s$ and $\boldsymbol{\xi}_t^h$. In this specific process, the linear transformation is firstly performed after splicing each vector vertically to convert it into a $d_k$ dimensional vector. Then calculate the weights of the features based on the normalized vector using the Softmax module. If the weight of a feature is close to 0, it means the importance of this feature is very low. Otherwise its importance is very high. The representation vectors are then updated in the following way,

$$
\begin{aligned}
\boldsymbol{\xi}^s &= \begin{bmatrix} \boldsymbol{\xi}_1^s \, \boldsymbol{\xi}_2^s \, \cdots \, \boldsymbol{\xi}_{m_1}^s \end{bmatrix} Softmax\Big( W_1 \begin{bmatrix} \boldsymbol{\xi}_1^{s\,T} \, \boldsymbol{\xi}_2^{s\,T} \, \cdots \, \boldsymbol{\xi}_{m_1}^{s\,T} \end{bmatrix}^T + \mathbf{b}_1 \Big), \\
\boldsymbol{\xi}_t^h &= \begin{bmatrix} \boldsymbol{\xi}_{t,1}^h \, \boldsymbol{\xi}_{t,2}^h \, \cdots \, \boldsymbol{\xi}_{t,m_2}^h \end{bmatrix} Softmax\Big( W_2 \begin{bmatrix} \boldsymbol{\xi}_{t,1}^{h\,T} \, \boldsymbol{\xi}_{t,2}^{h\,T} \, \cdots \, \boldsymbol{\xi}_{t,m_2}^{h\,T} \end{bmatrix}^T + \mathbf{b}_2 \Big),
\end{aligned}
\tag{2}
$$

where $\boldsymbol{\xi}^s \in \mathbb{R}^{d_k \times 1}$, $W_1 \in \mathbb{R}^{m_1 \times (d_k \cdot m_1)}$, $\mathbf{b}_1 \in \mathbb{R}^{m_1 \times 1}$, and $\boldsymbol{\xi}_{t,m}^h \in \mathbb{R}^{d_k \times 1}$, $W_2 \in \mathbb{R}^{m_2 \times (d_k \cdot m_2)}$, $\mathbf{b}_2 \in \mathbb{R}^{m_2 \times 1}$.

When processing the sequence features, since the subsequent structure of the algorithm includes the Attention module, it is necessary to assign positional encoding to the input vector, that is, add the temporal information of the $t^{th}$ moment to the sequence feature vector $\boldsymbol{\xi}_t^h$. Suppose that the temporal information at the $t^{th}$ moment is $TimeSeq_t$, where

$$TimeSeq_t = (t - p, t - p + 1, \ldots, t, \ldots, t + q).$$

In our application, the user life cycle sequence is a non-negative monotonic increasing sequence. For different products, users' payment habits show obvious personalized patterns. For example, the payment behavior is concentrated in the early or late stage of the customer's life cycle. Therefore, we hope to convert the $t^{th}$ moment's time tag information $TimeSeq_t$ into a representation vector of the feature sequence by the Embedding method, and then add it as a position code into $\boldsymbol{\xi}_t^h$. This is the major difference of processing static features and sequential features. The popular positional encoding composes of sine and cosine functions, which is not suitable here. In our TempInput algorithm the positional encoding is obtained via model training. The specific method uses the Embedding technique to convert the temporal label sequence $TimeSeq_t$ at the $t^{th}$ moment into a discrete feature vector, and then superimposes it into the output vector $\boldsymbol{\xi}_t^h$, that is

$$\boldsymbol{\xi}_t^h = \widetilde{\boldsymbol{\xi}}_t^h + Emb(TimeSeq_t),$$

where $\widetilde{\boldsymbol{\xi}}_t^h$ is the representation vector obtained from the previous step by equation 2. In addition, the position encoding also needs to be added on the Decoder side. The input of the Decoder module is denoted by $\boldsymbol{\xi}_t^f$ in the later sections.

---

**Algorithm 2:** StaInput

---

**Input:** The static feature data of user $\mathbf{x}^s = (x_1^s, x_2^s, \ldots, x_{m_1}^s)$
**Output:** $\boldsymbol{\xi}^s$ the embedding vector of static features

**Step 1:** For $i = 1 : m_1$, compute embedding vectors:

$$\boldsymbol{\xi}_i^s \leftarrow Emb_i(x_i^s)$$

**Step 2:** Take a weighted average of the embedding vectors:

$$\boldsymbol{\xi}^s = \begin{bmatrix} \boldsymbol{\xi}_1^s & \boldsymbol{\xi}_2^s & \cdots & \boldsymbol{\xi}_{m_1}^s \end{bmatrix} Softmax\left(W_1 \begin{bmatrix} \boldsymbol{\xi}_1^{s^T} & \boldsymbol{\xi}_2^{s^T} & \cdots & \boldsymbol{\xi}_{m_1}^{s^T} \end{bmatrix}^T + \mathbf{b}_1\right)$$

---

---

**Algorithm 3:** TempInput

---

**Input:** The sequence feature data of the user at time $t$ $\mathbf{x}_h^t = (x_{t,1}^h, x_{t,2}^h, \ldots, x_{t,m_2}^h)$,
   $\quad TimeSeq_t = (t - p, t - p + 1, \ldots, t, \ldots, t + q)$
**Output:** $\boldsymbol{\xi}_t^h$ the embedding vector of sequence features at time $t$

**Step 1: For** $j = 1 : m_2$, computing embedding vectors:

$$\boldsymbol{\xi}_{t,j}^h \leftarrow Emb_i(x_{t,j}^h)$$

**Step 2:** Take a weighted average of the embedding vectors:

$$\widetilde{\boldsymbol{\xi}}_t^h = \begin{bmatrix} \boldsymbol{\xi}_{t,1}^h & \boldsymbol{\xi}_{t,2}^h & \cdots & \boldsymbol{\xi}_{t,m_2}^h \end{bmatrix} Softmax\left(W_2 \begin{bmatrix} \boldsymbol{\xi}_{t,1}^{h^T} & \boldsymbol{\xi}_{t,2}^{h^T} & \cdots & \boldsymbol{\xi}_{t,m_2}^{h^T} \end{bmatrix}^T + \mathbf{b}_2\right)$$

**Step 3:** Convert the temporal label sequence $TimeSeq_t$ into a discrete feature vector, and add it to the representation vector:

$$\boldsymbol{\xi}_t^h = \widetilde{\boldsymbol{\xi}}_t^h + Emb(TimeSeq_t)$$

---

### A.2 LABEL LEARNING NETWORK

The Label Learning is designed to obtain the multi step predictions of the sequence label values.

### A.2.1 SEQ2SEQ MODULE

As shown in Figure 6, the Seq2seq module is composed of an Encoder module and a Decoder module. Each module corresponds to a LSTM cell and its subsequent network structure. Similar to the Representation Learning, the Seq2seq module also needs to deal with the static features and sequence features. In addition, we also need to add the position encoding at the input side of the Decoder. The encoded position vector is

$$\boldsymbol{\xi}_{t+j}^f = Emb(TimeSeq_{t+j}),$$

where $TimeSeq_{t+j} = (t - p, t - p + 1, \ldots, t, \ldots, t + j)$.

Suppose the representation vectors of the static and sequence features at time $t$ from the Representation Learning moduleis $\boldsymbol{\xi}^s$ and $\boldsymbol{\xi}_t^h$. We use $\boldsymbol{\xi}^s$ to initialize the hidden state $\boldsymbol{h}^{en}$ and the unit state $\boldsymbol{c}^{en}$ of the Encoder, where

$$\boldsymbol{h}_{t-p-1}^{en} = GLU(\boldsymbol{\xi}^s) = (W_3\boldsymbol{\xi}^s + b_3) \times \sigma(W_4\boldsymbol{\xi}^s + \mathbf{b}_4),$$
$$\boldsymbol{c}_{t-p-1}^{en} = GLU(\boldsymbol{\xi}^s) = (W_5\boldsymbol{\xi}^s + b_5) \times \sigma(W_6\boldsymbol{\xi}^s + \mathbf{b}_6),$$

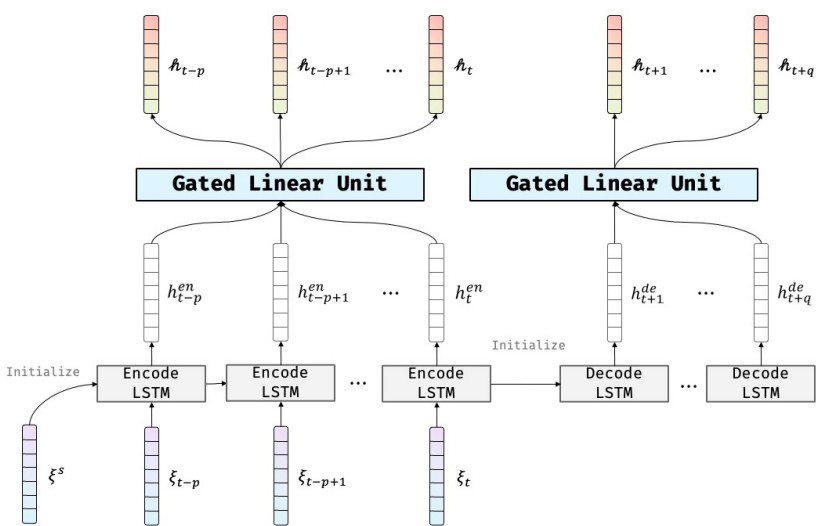

Figure 6: The operational structure of the Seq2seq module

where $\sigma(x) = \frac{1}{1+exp(-x)}$ is the sigmoid function. Then the Encoder model can calculate the hidden state of the current time $\boldsymbol{h}_t^{en}$ and the cell state $\boldsymbol{c}_t^{en}$. $\boldsymbol{h}_t^{en}$ is the output of the Encoder at time $t$. This process is different from the common Auto Encoder algorithm and the Seq2seq algorithm, where the common Seq2seq algorithm only cares about the last hidden state of the Encoder and the output of the Decoder. The main purpose of our Seq2seq module is to store the results of each step and pass them to the subsequent structures for further feature extractions of the timing characteristics of the data.

According to the model equation 1, the dimension of the input matrix of the Encoder is $(p+1) \times d_k$. The GLU (Gated Linear Unit) unit is a shallow back propagation neural network, which is often used for transition operations between the output and input of different structures. The computational process of the Encoder is

$$\{\boldsymbol{h}_{t-i}^{en}, \boldsymbol{c}_{t-i}^{en}\} = LSTM(\boldsymbol{\xi}_{t-i}^h, \boldsymbol{h}_{t-i-1}^{en}, \boldsymbol{c}_{t-i-1}^{en}),$$
$$\widetilde{\boldsymbol{h}}_{t-i} \leftarrow (W_7 \, \boldsymbol{h}_{t-i}^{en} + \mathbf{b}_7) \times \sigma(W_8 \, \boldsymbol{h}_{t-i}^{en} + \mathbf{b}_8),$$
$$\hbar_{t-i} \leftarrow \widetilde{\boldsymbol{h}}_{t-i} + W_{glu1} \, \widetilde{\boldsymbol{h}}_{t-i},$$

where $i = p, p-1, \ldots, 0$.

Similarly, on the Decoder side, the hidden state $h_t^{de}$ is the output of the Decoder at time $t$. The Decoder uses the last hidden state and unit state of the Encoder to initialize the hidden state and unit state. According to the model equation 1, the length of the input sequence at the Decoder side is $q$. The calculation process of the Decoder is

$$\boldsymbol{h}_t^{de} = \boldsymbol{h}_t^{en}, \quad \boldsymbol{c}_t^{de} = \boldsymbol{c}_t^{en}$$
$$\{\boldsymbol{h}_{t+j}^{de}, \boldsymbol{ch}_{t+j}^{de}\} = LSTM(\boldsymbol{\xi}_{t+j}^f, \boldsymbol{h}_{t+j-1}^{de}, \boldsymbol{c}_{t+j-1}^{de}),$$
$$\widetilde{\boldsymbol{h}}_{t+j} \leftarrow (W_9 \, \boldsymbol{h}_{t+j}^{de} + \mathbf{b}_9) \times \sigma(W_{10} \, \boldsymbol{h}_{t+j}^{de} + \mathbf{b}_{10})$$
$$\hbar_{t+j} \leftarrow \widetilde{\boldsymbol{h}}_{t+j} + W_{glu2} \, \widetilde{\boldsymbol{h}}_{t+j}$$

where $j = 1, 2, \ldots, q$.

404 Algorithm 4 is the computational workflow of our Seq2seq module. Our Seq2seq module adopts
405 some common structures, such as GLU and AddNorm, to improve the model training performance.
406 The AddNorm layer includes a Skip Connection structure (He et al., 2016) and a Batch Normal-
407 ization structure (Ioffe and Szegedy, 2015). The Skip Connection structure is used to reduce the
408 non-convexity of the network to protect the model from the gradient problem caused by the deep
409 learning structure. The Batch Normalization improves the training speed by normalizing each fea-
410 ture of samples within each batch. It is suitable for processing the structural data which has relatively
411 strong features independencies. The output of the Seq2seq module will be passed to the Attention
412 unit for the further analysis.

---

**Algorithm 4:** Seq2seq Module

**Input:** $\{\boldsymbol{\xi}^s, \boldsymbol{\xi}^h_{t-p}, \boldsymbol{\xi}^h_{t-p+1}, \dots \boldsymbol{\xi}^h_t\}$ the embeding vectors from the Representation Learning
**Output:** $\{\hbar_{t-p}, \hbar_{t-p+1}, \dots, \hbar_t, \dots, \hbar_{t+q}\}$

**The Encoder stage**: Initialize the hidden state and unit state of the Encoder LSTM $\boldsymbol{h}^{en}_{t-p-1}$
  and $\boldsymbol{c}^{en}_{t-p-1}$,

$$\boldsymbol{h}^{en}_{t-p-1} \leftarrow (W_3\,\boldsymbol{\xi}^s + b_3) \times \sigma(W_4\,\boldsymbol{\xi}^s + \mathbf{b}_4)$$
$$\boldsymbol{c}^{en}_{t-p-1} \leftarrow (W_5\,\boldsymbol{\xi}^s + b_5) \times \sigma(W_6\,\boldsymbol{\xi}^s + \mathbf{b}_6)$$

  **For** $i = p : 0$, compute:

$$\{\boldsymbol{h}^{en}_{t-i}, \boldsymbol{c}^{en}_{t-i}\} \leftarrow LSTM(\boldsymbol{\xi}^h_{t-i}, \boldsymbol{h}^{en}_{t-i-1}, \boldsymbol{c}^{en}_{t-i-1})$$
$$\widetilde{\boldsymbol{h}}_{t-i} \leftarrow (W_7\,\boldsymbol{h}^{en}_{t-i} + \mathbf{b}_7) \times \sigma(W_8\,\boldsymbol{h}^{en}_{t-i} + \mathbf{b}_8)$$
$$\hbar_{t-i} \leftarrow \widetilde{\boldsymbol{h}}_{t-i} + W_{glu1}\,\widetilde{\boldsymbol{h}}_{t-i}$$

**The Decoder stage**: Initialize the hidden state and unit state of the Decoder LSTM unit $\boldsymbol{h}^{de}_t$
  and $\boldsymbol{c}^{de}_t$,

$$\boldsymbol{h}^{de}_t \leftarrow \boldsymbol{h}^{en}_t, \boldsymbol{c}^{de}_t \leftarrow \boldsymbol{c}^{en}_t$$
$$\boldsymbol{\xi}^f_{t+j} = Emb(TimeSeq_{t+j})$$

  **For** $j = 1 : q$, compute:

$$\{\boldsymbol{h}^{de}_{t+j}, \boldsymbol{c}^{de}_{t+j}\} \leftarrow LSTM(\boldsymbol{\xi}^f_{t+j}, \boldsymbol{h}^{de}_{t+j-1}, \boldsymbol{c}^{de}_{t+j-1})$$
$$\widetilde{\boldsymbol{h}}_{t+j} \leftarrow (W_9\,\boldsymbol{h}^{de}_{t+j} + \mathbf{b}_9) \times \sigma(W_{10}\,\boldsymbol{h}^{de}_{t+j} + \mathbf{b}_{10})$$
$$\hbar_{t+j} \leftarrow \widetilde{\boldsymbol{h}}_{t+j} + W_{glu2}\,\widetilde{\boldsymbol{h}}_{t+j}$$

---

### A.2.2 ATTENTION MODULE

414 The Attention module analyze the sequence information processed by the Seq2seq module. Figure 7
415 is the operational structure diagram of the Attention module. First, the Attention module maps the
416 output of the Encoder-Decoder $\{\hbar_{t+i}| - p \le i \le q\}$ and the static feature vector $\boldsymbol{\xi}^s$ into the input
417 matrix of the Attention module, where the input matrix $M$ is defined as

$$M = \left[\mathbf{m}_{t-p}\,\mathbf{m}_{t-p+1}\,\dots\,\mathbf{m}_{t+q}\right]^T,$$
$$\mathbf{m}_{t+i} = W^{(1)}_{11}\hbar_{t+i} + W^{(2)}_{11}\boldsymbol{\xi}^s + \mathbf{b}_{11},$$

418 where $-p \le i \le q$. Since the proposed algorithm is composed of the self-attention unit, $M$ can be
419 used as the $K$ and $V$ matrices in the Attention layer, which means we can set $K = V = M$. For the
420 $q$-step prediction of time series problem studied in this article, we can only use the corresponding
421 input on the Encoder side as the source of $Q$ matrix, and then do the following calculation:

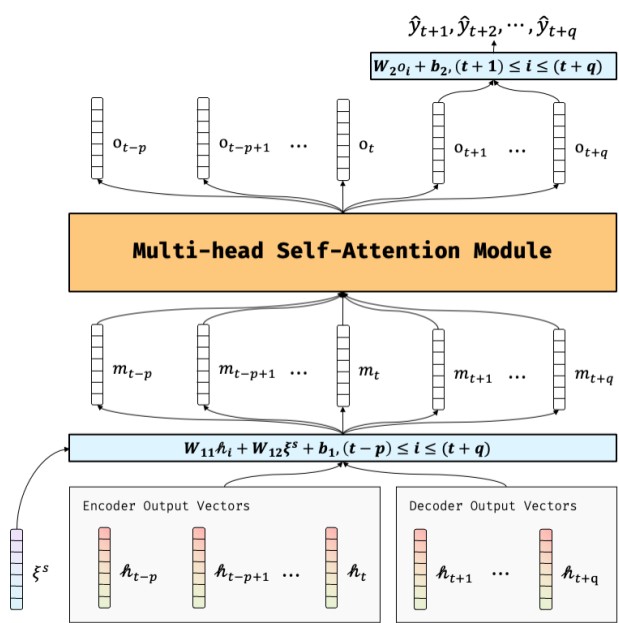

Figure 7: The operational structure of the Attention module

For the $q$-step predictions of the time series problem studied in this article, we only need to calculate the latent vectors from the moment $t + 1$ to $t + q$ through the Attention layer. Therefore $Q$ is constructed in the following way,

$$Q = \left[\mathbf{m}_{t+1}\, \mathbf{m}_{t+2}\, \ldots\, \mathbf{m}_{t+q}\right]^T.$$

In this case, the result of the Attention output is the hidden vectors from time $t + 1$ to $t + q$. We can also replace $Q$ by $M$, then the output of the Attention layer will be the latent vectors from time $t - p$ to $t + q$, which will be a waste of the computing resources since we only need the hidden vectors from time $t + 1$ to $t + q$. The final output of the Attention layer is the weighted average of the results of each self-attention head.

$$O_h \leftarrow Softmax(\frac{W_h^q\, Q\, (W_h^k\, K)^T}{\sqrt{d_k}})\, W_h^v\, V,$$

$$O \leftarrow \sum_{h=1}^{H} O_h, \quad O \in \mathbb{R}^{q \times d_k},$$

where $H$ is the number of heads of the Self-attention module, $W_h^q, W_h^k, W_h^v$ are the parameter matrices of each self-attention head. The output of the self-attention mechanism is also processed by the GLU and AddNorm units. These two operations do not change the dimension of the vectors. The output of the Attention layer is then processed by the Position-Wise Feed Forward Network Module to obtain the multi-step prediction vector $\mathbf{y}_t^{(q)}$ for the next $q$ moments in the future. The computational process is

$$\widetilde{O}_1 \leftarrow O + (OW_{12} + \mathbf{b}_{12}) \times \sigma(OW_{13} + \mathbf{b}_{13}),$$

$$\widetilde{O}_2 \leftarrow \left[\hbar_{t+1}\, \hbar_{t+2} \cdots \hbar_{t+q}\right]^T + (\widetilde{O}_1 W_{14} + \mathbf{b}_{14}) \times \sigma(\widetilde{O}_1 W_{15} + \mathbf{b}_{15}),$$

$$\mathbf{y}_t^{(q)} = W_{out}\widetilde{O}_2 + \mathbf{b}_{out}.$$

This network structure comes from the Transformer. It maps the high dimensional vector corresponding to each time step into a scalar through a shallow network. The scalar is the predicted label value. Algorithm 5 is the computational workflow of Attention module.

---

**Algorithm 5:** Attention Module

---

**Input:** The output vector of the Seq2seq module $(\hbar_{t-p}, \hbar_{t-p+1}, \ldots, \hbar_t, \ldots, \hbar_{t+q})$

**Output:** $\mathbf{y}_t^{(q)}$ the vector of the $q$-step predictions from moment $t+1$ to $t+q$

**For** i = -p: q, compute:

$$\mathbf{m}_{t+i} \leftarrow W_{11}^{(1)} \hbar_{t+i} + W_{11}^{(2)} \boldsymbol{\xi}^s + \mathbf{b}_{11}$$

the parameter matrix of the self-attention mechanism:

$$Q \leftarrow [\mathbf{m}_{t+1} \, \mathbf{m}_{t+2} \, \ldots \, \mathbf{m}_{t+q}]^T$$
$$K = V = M \leftarrow [\mathbf{m}_{t-p} \, \mathbf{m}_{t-p+1} \, \ldots \, \mathbf{m}_{t+q}]^T.$$

**For** h=1: H, compute:

$$O_h \leftarrow Softmax\left(\frac{W_h^q Q \, (W_h^k K)^T}{\sqrt{d_k}}\right) W_h^v V$$

Aggregate and compute the output of each attention mechanism head:

$$O \leftarrow \sum_{h=1}^{H} O_h$$
$$\widetilde{O}_1 \leftarrow O + (O \, W_{12} + \mathbf{b}_{12}) \times \sigma(O \, W_{13} + \mathbf{b}_{13})$$
$$\widetilde{O}_2 \leftarrow [\hbar_{t+1} \quad \hbar_{t-2} \quad \ldots \quad \hbar_{t+q}]^T + (\widetilde{O}_1 \, W_{14} + \mathbf{b}_{14}) \times \sigma(\widetilde{O}_1 \, W_{15} + \mathbf{b}_{15})$$

Predict the label value for the next $q$ steps in the sequence:

$$\mathbf{y}_t^{(q)} \leftarrow W_{out} \widetilde{O}_2 + \mathbf{b}_{out}$$

---

