# OpenReview forum: "Continuous Multi-step Predictions of Highly Imbalanced Multivariate Time Series via Deep Learning Network"
_ICLR.cc/2024/Conference — ICLR 2024 Conference Withdrawn Submission_

### Official Review · Reviewer_sXU7 · 2023-10-27

**Soundness:** 2 fair
**Presentation:** 2 fair
**Contribution:** 2 fair
**Rating:** 3
**Confidence:** 4

**Summary:**

The paper addresses the problem of zero inflation in time series
forecasting, i.e., to forecast a continuous-valued channel that
contains many zeros. The authors propose a method that
trains a recurrent neural network with an attention decoder
(for multiple timesteps) in two stages with two losses, a quantile
loss and a weighted quantile loss. In an experiment on a
public dataset they show that their method outperforms
a two stage approach from the literature, that first tries to
predict if the value is non-zero and then conditionally
on being non-zero the value.

**Strengths:**

s1. interesting problem: zero inflation, esp. in time series forecasting.

**Weaknesses:**

w1. aspects of the proposed method are not clear.
w2. experiments are run on a single dataset.
w3. the method is compared against a single, simple baseline.

**Questions:**

w1. aspects of the proposed method are not clear.
- how exactly is the weight w_i (p. 5) computed?
  - the "x" here have nothing to do with the covariates "x" in eq. 1, right?
  - why is "x_{(1)}" missing in line 148 ?
  - denotes "x_j" the same as "x^{(j)}" ?
  - will not always be exactly one "I(x_j \leq y_i < x_{j+1})" be
    one (for one j), and for all others be zero? So the weight is
    basically 1 / \int_{x_i}^{x_{i+1} f(x) dx ?
  - what is "f" ?

w2. experiments are run on a single dataset.
- a wider experimentation on more datasets will provide better
  evidence that the proposed method performs well.

w3. the method is compared against a single, simple baseline.
- As a decision tree ensemble, how is the baseline "2Stage-LGBM"
  treating the time series aspect of the problem?
  It will be more convincing to compare against a proper time
  series forecasting model.
- A simple, principled baseline is to decouple the target channel into
  two target channels: 1) the indicator for being zero and 2) the value,
  and then train any state-of-the-art forecasting model to forecast
  both channels at the same time (where the value channel is masked
  for zero values).
- A comparison against state of the art time series forecasting models
  such as Nie et al. 2023 and Zeng et al. 2023 is missing.

Writing needs some attention, e.g.,
- p. 1 "the prediction the user's payment values"

References:
- Nie, Yuqi, Nam H. Nguyen, Phanwadee Sinthong, and Jayant
  Kalagnanam. “A Time Series Is Worth 64 Words: Long-Term Forecasting
  with Transformers.”  ICLR 2023.
- Zeng, Ailing, Muxi Chen, Lei Zhang, and Qiang Xu. “Are Transformers
  Effective for Time Series Forecasting?” In AAAI, 2023.

---

### Official Review · Reviewer_BcAV · 2023-10-30

**Soundness:** 2 fair
**Presentation:** 1 poor
**Contribution:** 1 poor
**Rating:** 1
**Confidence:** 4

**Summary:**

**Summary**

The authors aim to address the data imbalance problem in multi-step time series forecasting by a joint model consisting of two parts: representation learning part and label learning part. They present experiments on a real-world dataset to show the effectiveness of the proposed method.

**Strengths:**

1. The research topic is interesting.

**Weaknesses:**

Majors:

1. The contribution of the paper is not well clarified.

2. For Step 1 Representation Learning, model structure details are missing. What is Emb() in Algorithms 2 and 3?

3. For Step 2 Label Learning, why choose LSTM, GLU, and multi-head self-attention? Why not use a Transformer-based model directly? The philosophy is unclear.

4. The authors introduce a Weighted Quantile Loss function, but no experiments are conducted to verify the effectiveness of this loss. Does it work better compared to the common Quantile Loss function or MAE?

5. The authors claim that FLIMTS has much less time cost compared to 2Stage-LGBM, but no experiments have been conducted to demonstrate that.

6. The dataset is split into a train set and a test set. As there is no validation set, how the hyperparameters are decided is not clear.

7. The authors claim that rAUC is a good metric for downstream tasks in reality, but the experiments show that FLIMTS does not have a significant advantage against 2Stage-LGBM.

8. Ablation studies and More methods for comparison are needed.

9. The content in the Appendix is more important than Fig. 2.

Minors:

10. I suggest the authors provide a citation and a full name for the dataset when it first appears in the paper. (AVSC first appears at Line 179, but is not explained until Line 185.)

11. Some citations are missing. For example, LSTM, GLU, and Multi-head Self Attention.

12. The authors use a date for the train/test split, but do not provide the starting/ending dates of the dataset.

13. At Line 87, the authors claim they do experiments on two datasets. However, in Section 3, there is only one dataset, AVSC.

14. There are some typos in the paper. For example,

  Line 31: the predicting the user’s payment values is a critical business requirement

  Line 71: It increases the model maintenance cost with a unsatisfactory prediction accuracy

  Line 94: where $\epsilon_t^{(q)}$ is the random error, $\mathcal{F}$ is a unknown nonlinear mapping function

**Questions:**

Please see the questions in the "Weaknesses" part.

---

### Official Review · Reviewer_wbNx · 2023-10-30

**Soundness:** 3 good
**Presentation:** 3 good
**Contribution:** 3 good
**Rating:** 5
**Confidence:** 4

**Summary:**

The paper proposes feature learning architecture FLIMTS, for imbalanced multivariate time-series which is a two-stage learning, (i) representation learning, which learns embedding for static and continuous features for multi-step prediction through quantile loss and (ii) label learning which learns weight for each prediction step of the quantile. FLIMTS has been adopted on AVSC shopped dataset with 11 transaction features and compared with an ML algorithm LightGBM.

**Strengths:**

1. FLIMT improvement is very high compared to LightGBM in terms of MSE, MAE, rAUC.
2. The proposed approach for multivariated imbalanced time-series data is novel and challenging.
3. FLIMTS seem to have better performance over LightGBM for different category imbalanced user dataset.

**Weaknesses:**

1. Comparison with some deep learning multivariate time-series architeture would have been better to analyze. E.g., Temporal Fusion Trasnformer (TFT), Deep AR are also based on multi-step quantile loss for multivariate time-series.
2. Fix some typos:
	- line 118: 'the' written twice
	- Author Contibution and acknowledgement section empty. You can remove these sections for submision pre-print.

**Questions:**

1. Would be convincing as well to see comparison with TFT on imbalance dataset. TFT is not a 2-stage learning but they also adopt self-attention mechanism and adapted for static and continuous time-series features.

2. Can you add some statistics, on overall for different category users how much FLIMTS have improved compared to baseline?

3. Overall paper and architecture is novel and impactful. However authors have compared with only one baseline. Is there no other baseline (specially deep learning-based) that can be slightly adapted and compared with such work? If TFT is not a comparable model, some explanation would also be easier to be convinced FLIMTS performance.

---

### Official Review · Reviewer_NGnU · 2023-11-01

**Soundness:** 2 fair
**Presentation:** 3 good
**Contribution:** 2 fair
**Rating:** 3
**Confidence:** 3

**Summary:**

In this work, the authors introduce a continuous multi-step prediction model for imbalanced multivariate time series data, which integrates a representation learning module, a label learning module, and a weighted quantile loss for imbalance issue. The model is evaluated against a basic machine learning model using real-world transaction datasets, demonstrating better performance.

**Strengths:**

The paper is well written, with clear experimental details.

**Weaknesses:**

My major comments are:

1. The selected baseline appears weak as it represents a basic machine learning model rather than comparative temporal deep learning models like GRU or LSTM. Moreover, the paper lacks ablation studies which would demonstrate the individual contributions of the different components of the proposed model. Without such experiments, it is difficult to evaluate the technical contributions of this work.

2. While data imbalance is positioned as a significant challenge, I didn't see the detailed statistics or plots showing the imbalance in the data. The authors claims that a 3:1 ratio is extremely imbalanced; however, in various prediction contexts, ratios of 10:1 or even 100:1 (positive to negative) are not uncommon. The authors should substantiate why this ratio brings significant challenges in their chosen domain.

3. What's the difference between the data imbalance issue here and the long-tail regression? There are many existing works addressing the long-tail regression issue. If relevant, please include these works in the discussion or baselines.

4. The paper mentions only a training set and a test set. How were the model's hyperparameters determined?

**Questions:**

Please address the weaknesses above.

---

> ### Comment · Reviewer_wbNx · 2023-11-20
>
> I second the weaknesses pointed out by reviewer Ngnu and bcav. I think the paper lacks several experiments:
> 1. It did not consider enough competitive baselines
> 2. Not enough metric
> 3. Only one dataset which is not enough to verify the soundness of proposed approach
>
> Hence, I changed my decision to weak reject from weak AC.

---

### Official Review · Reviewer_jHHC · 2023-11-14

**Soundness:** 2 fair
**Presentation:** 2 fair
**Contribution:** 2 fair
**Rating:** 5
**Confidence:** 3

**Summary:**

This paper focuses on data imbalanced problem in time-series problem. Authors introduce a novel model, named FLIMTS, incorporates a two-stage learning process: representation learning and label learning. The representation learning stage focuses on embedding static and continuous features for multi-step prediction using quantile loss, while the label learning stage assigns weights to each prediction step. The model also includes a weighted quantile loss to address the imbalance issue. FLIMTS has been tested against a basic machine learning model, LightGBM, using the AVSC shopped dataset, which includes 11 transaction features. The results demonstrate that FLIMTS outperforms the basic machine learning model, showcasing its effectiveness in handling multi-step time series forecasting in the presence of data imbalance.

**Strengths:**

1. Problem is important.
2. Proposed method is novel.
3. FLIMT seems to outperform LightGBM in various datasets.

**Weaknesses:**

1. The definition of 'Emb()' is missing in Algorithms 2 and 3.

2. There is a lack of both theoretical and empirical evidence supporting the superiority of the Weighted Quantile Loss.

3. The results for one of the datasets are missing, despite the authors' assertion of testing on two datasets (referenced in line 87).

4. Figure 5 should be relocated to the main body of the paper for better context and clarity.

5. Certain models and claims made by the authors require additional references or evidence for validation.

6. Typos present in the document need further correction.

**Questions:**

See above